# Metasurface-Based Quantum Searcher on a Silicon-On-Insulator Chip

**DOI:** 10.3390/mi13081204

**Published:** 2022-07-28

**Authors:** Zeyong Wei, Haoyu Li, Linyuan Dou, Lingyun Xie, Zhanshan Wang, Xinbin Cheng

**Affiliations:** 1Institute of Precision Optical Engineering, School of Physics Science and Engineering, Tongji University, Shanghai 200092, China; weizeyong@tongji.edu.cn (Z.W.); 1852468@tongji.edu.cn (H.L.); 1852053@tongji.edu.cn (L.D.); wangzs@tongji.edu.cn (Z.W.); chengxb@tongji.edu.cn (X.C.); 2MOE Key Laboratory of Advanced Micro-Structured Materials, Tongji University, Shanghai 200092, China; 3Shanghai Frontiers Science Research Base of Digital Optics, Tongji University, Shanghai 200092, China

**Keywords:** metasurface, on-chip, optical analog computing, quantum search algorithm

## Abstract

Optical analog computing has natural advantages of parallel computation, high speed and low energy consumption over traditional digital computing. To date, research in the field of on-chip optical analog computing has mainly focused on classical mathematical operations. Despite the advantages of quantum computing, on-chip quantum analog devices based on metasurfaces have not been demonstrated so far. In this work, based on a silicon-on-insulator (SOI) platform, we illustrated an on-chip quantum searcher with a characteristic size of 60 × 20 μm^2^. We applied classical waves to simulate the quantum search algorithm based on the superposition principle and interference effect, while combining it with an on-chip metasurface to realize modulation capability. The marked items are found when the incident waves are focused on the marked positions, which is precisely the same as the efficiency of the quantum search algorithm. The proposed on-chip quantum searcher facilitates the miniaturization and integration of wave-based signal processing systems.

## 1. Introduction

Exploring new approaches to improve the computational power and efficiency is a destination that researchers have been constantly pursuing. Digital computers generally are inefficient and suffer from high power consumption when solving specialized computational assignments, for instance, imaging processing and edge detection [1,2]. Optical analog computing, however, has attracted interest owning to its natural advantages of the capability of parallel processing, high speed and low power consumption compared to traditional digital computing [3].

Since the idea of the computational metamaterials was first proposed by Silva [4], numerous spatial-domain devices have been illustrated, such as differentiators [5,6], integrators [7,8], convolvers [9] and equation solvers [10]. However, most of the metasurface-based optical computing systems are designed in free space [11,12,13,14]. They are still bulky and difficult to integrate, although the sizes of them have been reduced by several orders of magnitude compared to the conventional 4-F systems [15,16,17]. Therefore, integrated analog optical computing systems at the chip level are extremely attractive [18,19]. Researchers have demonstrated some preparation schemes for on-chip analog optical computing devices, such as on-chip optical differentiators [20], integrators [21], convolvers [22], and optical diffractive neural networks [23,24]. Integrated photonics has shown extraordinary potential in optical computing [25].

Optical quantum computing is an attractive route toward quantum supremacy [26]. The previous studies about on-chip optical analog computing are mainly focused on classical mathematical operations. However, quantum computation greatly outperforms classical computing in terms of computation speed and is expected to play an important role in future information processing [27]. In addition, some fundamental properties of quantum computing, including the superposition principle and interference phenomena, are common to both classical waves and quantum mechanics [28,29]. Therefore, it is possible to simulate quantum algorithms with classical waves. In quantum algorithms, the quantum search algorithm is extremely important. Rapid searching is usually central to solving some difficult problems. In the search process, the quantum search algorithm is more efficient than the classical one in general. To date, the quantum search algorithm has been implemented under numerous models, such as optical experiments [30], NMR systems [31,32], ion traps [33] and metamaterials [34]. Unfortunately, an on-chip quantum searcher based on a silicon metasurface is still to be investigated.

In this work, we proposed an on-chip quantum searcher based on the silicon-on-insulator (SOI) platform. SOI consists of a substrate made up of silicon (Si), on top of which there is a sub-micrometer-thick Si film on a silica (SiO_2_) buried layer. We simulated the quantum search algorithm with classical waves while combining it with the modulation capability of the on-chip metasurfaces to implement the on-chip quantum search algorithm. By engineering the width and length of each slot on a SOI chip, the control of the transmitted intensity and phase profile can be realized. The designed on-chip quantum searcher is comprised of four on-chip metasurfaces. The whole device has a compact size, with a length and width of 60 µm and 20 µm, respectively. In a proof-of-concept demonstration, the output distribution fits well with the effective medium system under the incidence of a representative light field at 1550 nm with transverse electric (TE) mode (polarized in y-direction), which demonstrates the feasibility of the proposed scheme. The on-chip quantum searcher proposed here facilitates the miniaturization and integration of communication, wave-based analog computing and signal processing systems.

## 2. Materials and Methods

The schematic diagram of proposed on-chip quantum searcher is shown in Figure 1. The incident signal *f*(*y*) is in the input plane of a 4-F system composed of four specially designed metasurfaces, containing an oracle metasurface, two identical metalenses and a middle metasurface (MMS). The first specially designed metasurface is placed at the incident light to mark the searched term. Two identical metalenses are used to perform the Fourier transforms in the 4-F system, and another metasurface is embedded in the middle of 4-F system to perform the quantum search operation in the Fourier plane.

In this work, we used the commercial software from Lumerical for simulation. The on-chip quantum searcher was designed and simulated using the finite difference time domain (FDTD) method. The simulated area is a three-dimensional (3D) rectangular region, which is nonuniformly gridded into slight meshes. The minimum mesh step is set to 50 nm, and the simulation time is set to 2000 fs so that the simulation results can be accurate. The perfectly matched layer-absorbing (PML) boundary conditions are applied in all directions in the simulation. At 1550 nm, the refractive indices of Si and SiO_2_ are 3.48 and 1.44, respectively. The incident light is set to TE mode light along the positive x-direction (polarized in y-axis). To realize the proposed scheme in a relatively easy-to-fabricate way, we designed the quantum searcher based on the SOI platform, which contains rectangular etched slot arrays (Figure 1). The entire device is made up of 4 metasurfaces consisting of 41 etched slots each. In this research, one slot is designed in the middle of each unit of the metasurface in the Si layer with a thickness of *h* = 250 nm. The center spacing of the adjacent units (*a* = 500 nm) is set to less than half of the wavelength, so that the total width of the device is within 20 μm. A rectangular slot is etched in the middle of each unit to form an on-chip rectangular waveguide. As shown in Figure 1, the width of the slot in each unit is *w*. On the SOI chip, the phase will be delayed as the wave passes through the slots. The phase is mainly determined by the length of the slots, so that the phase can be modulated by changing the length of the slots. Numerical simulation demonstrates that the thickness of the upper Si layer should be determined as *h* = 0.25 μm and the width of the rectangular waveguide as *w* = 140 nm to ensure efficient transmission of wave and effective phase modulation. 

In order to realize on-chip quantum search, the oracle metasurface is designed to realize a phase shift $\phi_{1}$ of 0 and *π*/2 depending on the value of the spatial position “*y*” as shown in the following equation: (1)$$\phi_{1}\left(y\right)=\left\{\begin{array}{c} \;0.5\pi ,\;\;y_{1}<y<y_{2}\\ \;0\pi ,\;\;\;\;\;\;\;\;\;\;\;\;\;\;\;\;\;else, \end{array}\right.$$
where from *y*_1_ to *y*_2_ denotes the position of the marked items. When the incident wave is illuminated on the oracle metasurface, the phase of the transmitted wave will be modulated and the incident beam will be imprinted with a spatially dependent phase profile to mark the target item. With the first metalens performing Fourier transform to the input signal, the distribution of spatial frequency is presented at the Fourier plane, where the middle metasurface (MMS) is placed between the two metalenses for phase modulation. The MMS, just like the oracle metasurface, is designed to realize a phase shift $\phi_{2}$ of 0 and *π*/2 depending on the value of the spatial position “*y*” as described in the equation below:(2)$$\phi_{2}\left(y\right)=\left\{\begin{array}{c} \;0.5\pi ,\;\;\;\left|y\right|<\left(y_{2}-y_{1}\right)/2\\ \;0\pi ,\;\;\;\;\;\;\;\;\;\;\;\;\;\;\;\;\;\;\;\;\;\;\;else, \end{array}\right.$$
where from *y*_1_ to *y*_2_ denotes the position of the marked items. The MMS can be performed by a properly designed metasurface as the oracle metasurface, except that the items with 0.5*π* phase transitions are moved to the position around *y* = 0 µm. The manipulated light pattern is then converted back to spatial domain with the assistance of the second metalens. The combination of two Fourier transform metasurfaces and the MMS can convert the phase difference marked by the oracle subblock into amplitude information by the sequences metalens-MMS-metalens, which is similar to the “inversion-about-average (IAA)” operation of the original quantum search algorithm [35]. After several searches, the on-chip wave can be completely concentrated on the searched positions in theory, which means the marked items are proven to be found. In this work, the oracle metasurface and the MMS are designed by varying the length of the etched slots in different units to satisfy the modulation requirements given by Equations (1) and (2).

It is worth mentioning that in classical wave simulation, some important factors introduced by the optical system need to be considered. Otherwise, they may influence the accuracy of the final results, such as aberrations, diffraction and the impedance mismatch [28]. As shown in the insets of Figure 1, in order to reduce the effect of optical diffraction, the length of slots is varied gradually between the position of the marked items (0.5*π* phase) and the surrounding areas (0 phase). It achieves a transition from 0.5*π* to 0 phase and avoids the undesirable effects caused by abrupt phase changes.

To implement the spatial Fourier transform in the on-chip 4-F system, we used two metalenses for on-chip beam control. The metalenses consist of an on-chip array of rectangular etched slots with varying length. Along the y-direction, they can impose a space-dependent phase shift on the incident light (TE polarized). The phase shift $\phi_{3}$ can be defined by the following formula [20]: (3)$$\phi_{3}\left(y\right)=\frac{2\pi }{\lambda_{0}}n_{eff}\left(f-\sqrt{f^{2}+y^{2}}\right),$$
where *λ*_0_ is the design wavelength in free space, *n_eff_* is the effective refractive index of the light guide confined in the silicon slab, and *f* is the focal length.

Among the previously reported metalenses, some bring different phases by different sizes and configuration structures; some achieve this by changing the alignment angle of a similar configuration; and others use different lengths of slots to obtain the desired wavefront. In this work, we obtain our desired phase shift by adjusting the length and width of the etched slots in each unit of the metalenses. These two metalenses and the two identical metasurfaces mentioned earlier are combined to form an on-chip 4-F system.

## 3. Results

The design of the on-chip quantum searcher is roughly divided into three steps. First, the phase and transmittance of the slot waveguide in a single unit are analyzed to build a database by using simulation calculations. Then, the design of device is carried out on the basis of the database. Finally, the device function of the quantum search is obtained and demonstrated.

### 3.1. Unit Analysis

The proposed metasurfaces consisting of rectangular etched slot arrays are based on the silicon-on-insulator (SOI) platform (Figure 2a). In this research, in order to design the beam modulation devices with superior performance, we first swept the parameters of the unit structure with the simulation software Lumerical FDTD Solutions. The thickness of the etched slots was determined as *h* = 250 nm and the period width of the unit was set as *a* = 500 nm. The periodic boundary was applied in the y-direction, while the perfectly matched layer-absorbing (PML) boundary was applied in the x- and z-directions to carry out the simulation calculations. The cross section of a single unit is shown in Figure 2b.

The large refractive index contrast between silicon and silicon dioxide (>2) allows a phase shift of 2*π* to be realized with high transmission [36]. Based on the above parameters, the length (*L*) and width (*w*) of the etched slot were swept to obtain the transmission of the metasurface. The results are shown in Figure 2b. It is obvious that the transmission of the metasurface varies greatly with the increase in the etched slot width. When the slot had a width *w* = 140 nm, the transmission of the unit structure could be maintained above 80%, which is suitable as the basic structure of the on-chip metasurface device. Therefore, the width of the etched slot waveguide was determined as *w* = 140 nm. Based on this, the phase with length L varying was simulated. The results are shown in Figure 2c. Varying the length from 0.2 µm to 2.5 µm, the phase of transmitted waves can cover the range from 0 to 2*π* while the transmission maintains greater than 80% simultaneously. 

### 3.2. Design of the Devices

The design of the SOI on-chip device was performed with the help of numerical results of unit structure parameters sweeping. Based on the attained database, the slot width (*w* = 140 nm) was determined to ensure a high transmission. PML boundary conditions were applied in all directions in the simulation.

In this research, we used the designed metasurfaces, oracle metasurface and the MMS to implement the on-chip phase modulation. The metasurfaces consist of a gradient varying on-chip transmit array. The corresponding length of on-chip etch slots was obtained according to the phase distributions given in Equations (1) and (2), in this research, *y_1_* = −2.75 µm and *y_2_* = −1.25 µm. The length of the etched slots at the position with 0.5π phase transition (from 1.25 μm to 2.75 μm) is 2 μm, and the length of the etched slots at the position with 0 phase transition is 2.5 μm. In order to reduce the influence of the optical diffraction, the length is varied gradually (2.08, 2.15, 2.24, 2.32, and 2.41, units: μm) between the positions of the marked items and surrounding areas. The structures of the oracle metasurface and the MMS are shown in Figure 3a,b. Figure 3c,d show the corresponding light-intensity distribution in the x-y plane in the middle of the 250 nm-thick silicon plate (at *z* = 0). The designed metasurfaces are both 20 µm wide in the y-direction.

In this research, we used the proposed metalens to implement the spatial Fourier transform. The metalens is made up of a gradient-varying on-chip transmit array [20], which along the y-direction imposes a space-dependent phase shift on the incident light (TE polarized) along the x-direction. According to phase derived from Equation (3), we then obtained the corresponding length of on-chip etched slot. The structure of the metalens is shown in Figure 4a. In Figure 4b, we demonstrate the electric-intensity distribution in the x-y plane in the middle of a 250 nm-thick silicon slab (at *z* = 0). The wavelength of the incident light is 1550 nm and the designed metalens is 20 µm wide in the y-direction with a focal length of 20 µm. Figure 4c, d show the detailed in-plane electric-field distribution of *|E_y_|* and *E_y_* across the metalens, respectively.

### 3.3. Result of Quantum Search

In order to test the efficiency of the on-chip quantum searcher, we performed numerical simulations using the finite difference time domain method (Lumerical FDTD Solutions). Using the previously mentioned metalens, oracle metasurface and the MMS, we built an on-chip 4-F system to perform the quantum search. PML boundary conditions were applied in all directions in the simulation. Figure 5a shows the snapshot of the normalized electric-field intensity for the entire on-chip system with the input function *E(y)* = *exp*(−*y*^2^/20). In Figure 5b, we also plot the distribution of the normalized electric-field intensity |*E(y)*|^2^ at the output cross section for 0.5-iterations. It could be found that the incident waves are mainly concentrated around the marked position. However, still some waves are not concentrated on the searched positions. As the number of iterations increase, the concentration of the electric field will become more significant. This corresponds to an increasing probability (approach to unity) of the searched states with the searching time increases. 

In order to further verify the accuracy of our designed on-chip quantum searcher, we changed the position of the marked position (change the 0.5*π* phase shift position, *y_1_* = 1.25 µm and *y_2_* = 2.75 µm). The snapshot of the normalized electric-field intensity of the whole device (change the position of the marked items) is shown in Figure 5c. The normalized output profile is shown in Figure 5d. In addition, the designed device can also be searched for multiple items when the oracle metasurface has more than one marked position. Our search results are similar to those of the metamaterial-based quantum searcher, which are given in the Appendix A.

## 4. Discussion

Thus far, we have designed an on-chip quantum searcher based on the SOI platform. By designing the width and length of each slot on SOI platform, the control over the transmission and phase shift can be realized. The center spacing of the adjacent units (*a* = 500 nm) and the width of etched slot (*w* = 140 nm) are fixed to ensure a high transmission. The slot length varies along the y-direction to achieve the desired wavefront. Based on the excellent phase-modulation capability of the on-chip metasurfaces, we demonstrated the quantum search algorithm in an on-chip 4-F system using four metasurfaces. We used the different degrees of freedom of photons to encode quantum information. By exploiting the common properties of classical optics and quantum mechanics, such as superposition and interference, we simulated the quantum search algorithm with classical light. The size of this quantum searcher is 60 × 20 µm^2^. Our numerical simulations clearly confirm that the search efficiency of the on-chip quantum searcher is the same as that of quantum computing and can search the marked items extremely significantly.

Furthermore, the general design principles in our device can be applied to any wavelength. In addition to the implementation of quantum search algorithms, other quantum algorithms can be simulated in a similar approach, such as the Deutsch–Jozsa algorithm. The on-chip quantum search simulator proposed in this work provides a novel approach for implementing quantum algorithms. It enriches and improves the wave beam modulation capability of on-chip metasurfaces and fills the gap in the field of quantum computing for on-chip optical computing devices. Our on-chip quantum searcher facilitates the miniaturization and integration of wave-based signal processing systems.

## Figures and Tables

**Figure 1 micromachines-13-01204-f001:**
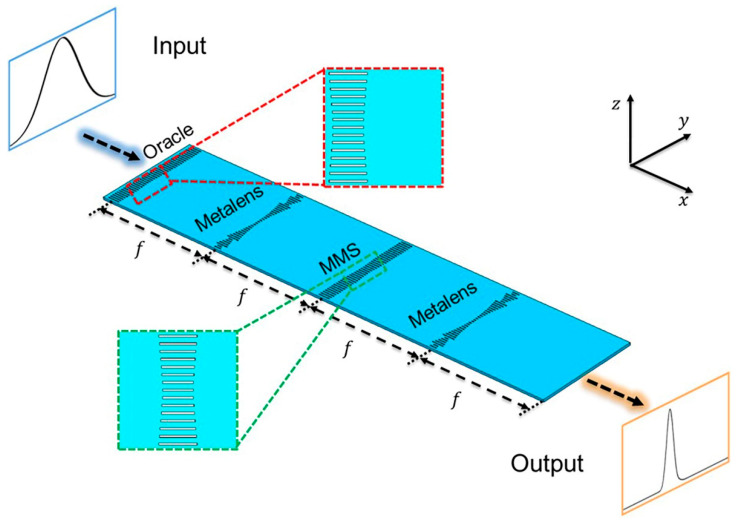
Schematic diagram of the proposed quantum searcher. The designed quantum searcher is comprised of four metasurfaces: an oracle metasurface, two metalenses, and a middle metasurface (MMS). Two insets: the corresponding on-chip structures. The term *f* = 20 µm denote the focal length of the metalens. The TE wave of wavelength *λ* = 1550 nm travels in the positive x-direction (polarized in the y-direction).

**Figure 2 micromachines-13-01204-f002:**
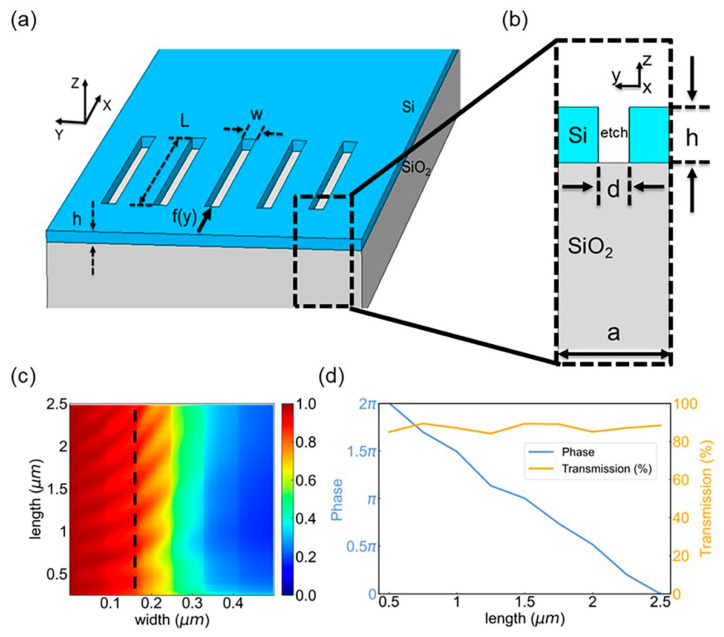
The design of the on-chip unit structure. (**a**) Schematic diagram of the three-dimensional structure of the on-chip metasurface-based device with wavelength *λ* = 1550 nm. *w* and *L* are the width and length of the etched slots, respectively. h is thickness of the Si layer, and *f*(*y*) is the incident signal. (**b**) Schematic diagram of the cross section of the device unit structure. The term *h* = 0.25 μm denotes the thickness of the upper Si layer of the SOI and the etched slots. The slots are etched in the middle of the upper Si layer. (**c**) Calculated transmission results. The length of etched slot waveguide varies from 0.2 μm to 2.5 μm and the width varies from 0 to 0.5 μm. The simulated transmission is a function of the width and length of the slot. The wavelength of the input light is 1550 nm (polarized in y-direction). (**d**) Calculated phase and transmission results. The simulated transmission and phase are related to the length of the slot. The length of the etched slot waveguide varies from 0.2 μm to 2.5 μm while the width of etched slot is fixed at *w* = 140 μm. The slot waveguide with a length of 0.5 μm to 2.5 μm brings phase coverage from 0 to 2*π*, while the transmittance is more than 80%.

**Figure 3 micromachines-13-01204-f003:**
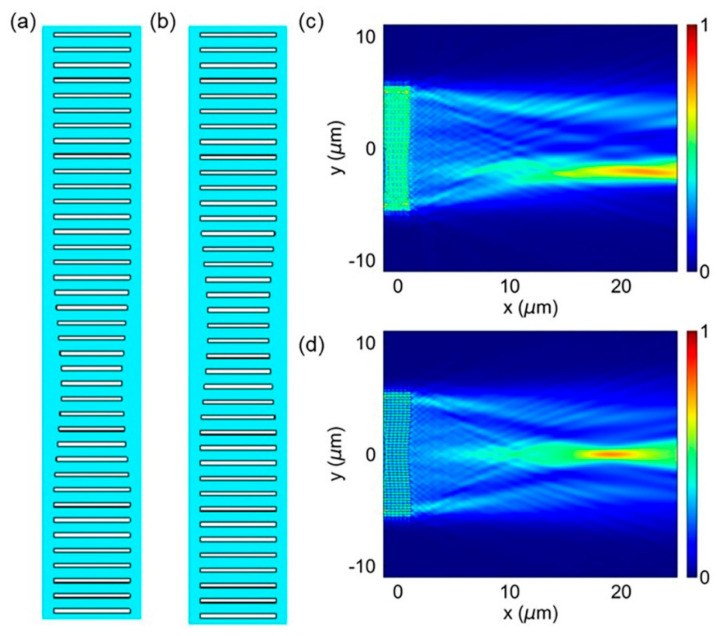
The design of on-chip oracle metasurface and MMS. (**a**) The structure diagram of the proposed oracle metasurface. (**b**) The structure diagram of the proposed MMS. (**c**,**d**) Simulation results for the in-plane wave distribution of |E_y_|^2^ in the middle of the silicon slab with incident light parallel to its optical axis are applied: (**c**) oracle metasurface, (**d**) MMS.

**Figure 4 micromachines-13-01204-f004:**
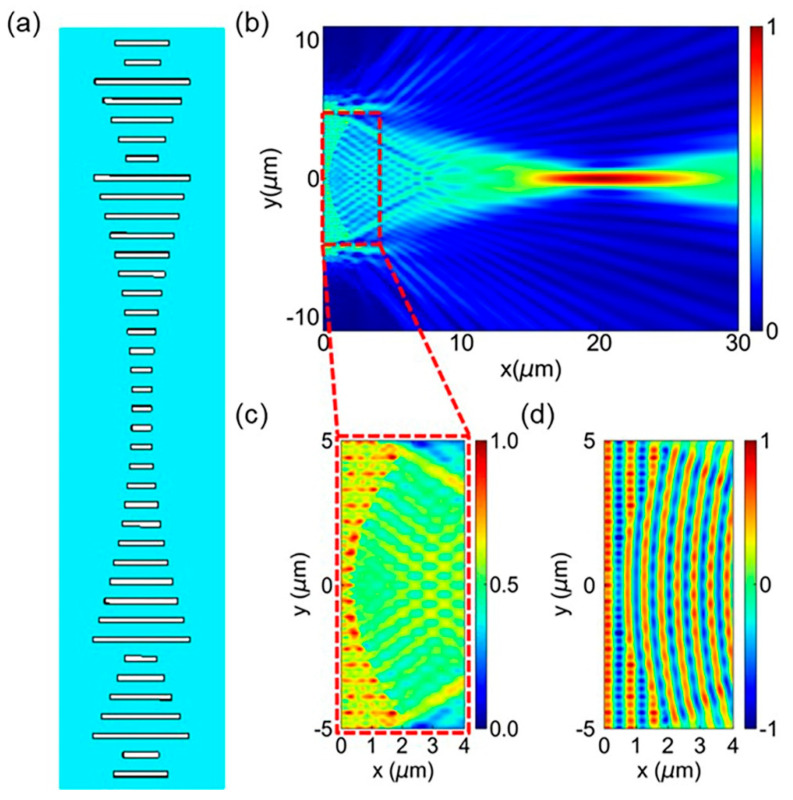
The design of on-chip metalens. (**a**) The structure diagram of the proposed metalens. (**b**) In-plane electric-field distribution |*E_y_*| in the middle of the silicon slab with incident light parallel to its optical axis. (**c**,**d**) Simulated electric-field distribution |*E_y_*| and *E_y_* in the region highlighted by a dashed red box in (**b**), respectively.

**Figure 5 micromachines-13-01204-f005:**
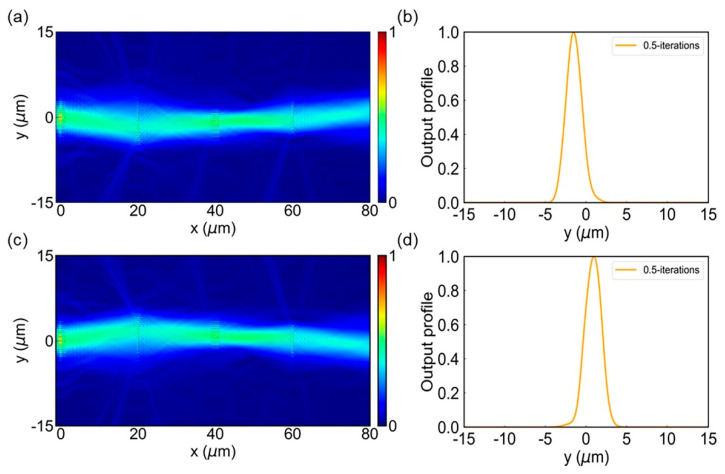
Theoretical simulation of the on-chip quantum search algorithm. (**a**,**c**) The snapshots for the intensity of the incident microwave throughout the metamaterial functioning as quantum searching simulator with 0.5-iterations: (**a**) *y_1_* = −2.75 µm and *y_2_* = −1.25 µm, (**c**) *y_1_* = 1.25 µm and *y_2_* = 2.75 µm. (**b**,**d**) Simulation results for the normalized output intensity of the metamaterial with the incident wave propagating 0.5-roundtrips within the device: (**b**) *y_1_* = −2.75 µm and *y_2_* = −1.25 µm, (**d**) *y_1_* = 1.25 µm and *y_2_* = 2.75 µm.

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
