# Peer review of "Metasurface-Based Quantum Searcher on a Silicon-On-Insulator Chip"

_micromachines, 2022, doi:10.3390/mi13081204_

Round 1

Reviewer 1 Report

The authors have demonstrated an on-chip quantum searcher based on the SOI platform. The size of the proposed device, 60 × 20 µm2, is several orders of magnitude smaller compared to the conventional quantum searchers. This manuscript is quite interesting. The numerical results nicely demonstrate their idea. I recommend the publication in Micromachines with some minor issues listed for the authors:

1) Please emphasize the novelty of the proposed device structure.

2) Although some references are introduced to show that it is possible to simulate quantum algorithms with classical waves, some additional discussions or explanations are still helpful for the reader.

3) What do w, L, h and f(y) mean in the Figure 2a?

Reviewer 2 Report

In this paper, the authors propose a design of SOI metasurface with the function of on-chip quantum search. The device realized beam modulation by setting equal-width etch slots on the Si layer to form metasurfaces. Comprehensive simulations are given to demonstrate the quantum search capability of the SOI chip. This work provides a novel strategy for on-chip quantum algorithms. I recommend the publication in Micromachines. However, the following issues should be addressed before it is accepted:

 1) Please enrich description of the proposed device structure.

 2) The schematic diagram of the three-dimensional structure of the on-chip device given in Figure 2a is a bit skewed, please draw it correctly.

 3) Please verify the FDTD simulator mentioned in the supplementary material (or describe model calibration).

 4) Citations of the relevant articles on optical computing can add to the broad interests of this work.
